# Predicting Extent of Resection and Neurological Outcome for Insular Gliomas: An Analysis of Two Available Classifications

**DOI:** 10.3390/cancers16244137

**Published:** 2024-12-11

**Authors:** Francesco Guerrini, Viola Marta Custodi, Antonio Giuri, Maria Claudia Caporrimo, Paola Bini, Ilaria Imarisio, Sara Colombo, Elisabetta Bonzano, Paolo Pedrazzoli, Enrico Marchioni, Luisa Chiapparini, Giannantonio Spena

**Affiliations:** 1Unit of Neurosurgery, Department of Head & Neck Surgery, Fondazione IRCCS Policlinico San Matteo, 27100 Pavia, Italy; f.guerrini@smatteo.pv.it (F.G.); v.custodi@smatteo.pv.it (V.M.C.); 2Unit of Neuroradiology, Department of Imaging Diagnostic, Fondazione IRCCS Policlinico San Matteo, 27100 Pavia, Italy; a.giuri@smatteo.pv.it (A.G.); m.caporrimo@smatteo.pv.it (M.C.C.); l.chiapparini@smatteo.pv.it (L.C.); 3Unit of Neurooncology, Fondazione IRCCS Mondino, 27100 Pavia, Italy; paola.bini@mondino.it (P.B.); enrico.marchioni@mondino.it (E.M.); 4Unit of Oncology, Department of Oncology, Fondazione IRCCS Policlinico San Matteo, 27100 Pavia, Italy; i.imarisio@smatteo.pv.it (I.I.); p.pedrazzoli@smatteo.pv.it (P.P.); 5Unit of Radiotherapy, Department of Oncology, Fondazione IRCCS Policlinico San Matteo, 27100 Pavia, Italy; s.colombo@smatteo.pv.it (S.C.); e.bonzano@smatteo.pv.it (E.B.)

**Keywords:** insula, glioma, Berger–Sanai, Kawaguchi

## Abstract

Berger–Sanai is a well-known insular glioma classification based on tumor topography; on the contrary, Kawaguchi’s classification considers the radiological attitudes and anatomical factors of gliomas. This study aimed to evaluate and compare them in terms of extent of resection and clinical outcome, both at short- and long-term follow-up. Berger–Sanai was found to be associated with the extent of resection and to have a moderate to strong power in predicting it. Both classifications were associated with postoperative clinical outcome, but this correlation was lost at 6 months. Only the Kawaguchi system was related to IONM changes and postoperative ischemia occurrence.

## 1. Introduction

Deep within the Sylvian fissure, the insula is a part of the paralimbic cortex, which plays a role in various autonomic and behavioral functions. It has close connections to the basal ganglia and key white matter tracts, including the corticospinal tract (CST), arcuate fasciculus (AF), uncinate fasciculus (UF), and inferior fronto-occipital fasciculus (IFOF). Insular gliomas are relatively rare, making up about 25% of all low-grade lesions. However, their surgical removal poses a significant challenge due to their proximity to these critical structures [1,2,3]. Furthermore, the lenticulostriate and long insular arteries are at risk during surgery, with the potential for severe neurological consequences if injured [4,5,6]. The complexity of these tumors is further compounded by their heterogeneity in terms of biological behavior (e.g., aggressiveness and growth rate) and morphological features (e.g., volume, and the tendency to infiltrate, compress, or displace delicate structures), making surgical decisions even more difficult. As with other gliomas, the principles of safe maximal resection apply to insular gliomas. However, due to the complex anatomy of the region and the risk of significant functional impairment, predicting morbidity in relation to extensive resections remains a key challenge in this surgery. Currently, the Berger–Sanai four-quadrant system (B-S) is a well-established classification method for insular gliomas. Based on topographic criteria, it divides tumor distribution along two planes intersecting at the foramen of Monro and the Sylvian fissure [7,8,9]. Similarly, Kawaguchi et al. introduced a scoring system that considers biological and morphological factors, such as tumor edges, contrast enhancement, involvement of the superior insular sulcus, and relationship with the lenticulostriate arteries (LSAs) [10]. Higher scores in this system are associated with a greater extent of resection. Despite being based on different parameters, both classification systems have shown good prognostic value regarding morbidity and extent of resection. In this study, we examined our experience with surgically treated insular gliomas, focusing on the use of these classification systems. Our hypothesis is that, due to their inherent differences, these systems may perform better in evaluating different outcome measures.

## 2. Materials and Methods

All patients surgically treated for insular glioma between 2015 and 2023 were included in this study. Preoperative clinical data included age, sex, onset symptoms, neurological examination, and Karnofsky Performance Status (KPS).

All patients underwent preoperative, postoperative (not beyond 48 h), and follow-up MRIs at our Neuroradiology Department. A 1.5 T MRI system was used, and the following sequences were performed during the same scan: T1-weighted images before and after gadolinium administration, 0.6 mm thickness FLAIR, and T2-weighted images. Diffusion-weighted imaging (DWI) sequences followed a routine clinical protocol using a standard echo-planar imaging (EPI) sequence, and ADC maps were automatically calculated. Postoperative MRI was performed within 48 h of surgery.

Tumor volume was calculated using T1-weighted sequences with gadolinium if contrast enhancement was present; for suspected low-grade gliomas, volume was determined using FLAIR sequences. Lesions were classified according to the B-S and Kawaguchi systems following the authors’ guidelines.

EOR was determined by postoperative MRI. Gross total resection (GTR) was defined as ≥99% of preoperative contrast-enhanced or hyperintense FLAIR volume removed. Subtotal (STR) or partial (PTR) resections were defined as 90–98% and <90% volume reduction, respectively.

Postoperative clinical status was evaluated immediately after surgery and at 6-month follow-up. A “good outcome” was defined as clinical stability or improvement, while worsening (transient or permanent at 6 months) was considered a “bad outcome”.

Pathological diagnosis was made according to the 2016 and 2021 WHO classification systems. Data collection was approved by the local Ethics Committee [11,12].

*Intraoperative protocol*. Patients were operated on through extensive intraoperative neurophysiological monitoring (IONM) (EEG, EMG, trancranial MEPs, cortical strip MEPs, subcortical direct stimulation of the corticospinal tract). In some cases, we preferred to utilize awake surgery protocol. The choice for an awake surgery over a IONM was mainly based on the following criteria: if the tumor was located in the dominant hemisphere and infiltrating deep white matter fibers; if the tumor was infiltrating the frontal opercula (in both dominant and non-dominant hemispheres); or if the tumor was in the dominant hemisphere but confined to the insula, we still operated with the aim of IONM alone. Awake surgery was performed employing an asleep–awake–asleep technique, beginning with a scalp block by mepivacaine 1% and ropivacaine 7% with 5 mcg/mL epinephrine; then conscious sedation with 0.3–0.8 mcg/kg/h dexmetedodimine and remifentanil 0.02–0.05 mcg/kg/min was performed (asleep phase); next, drug infusion was strongly reduced after dural opening to permit cortical and subcortical mapping (awake phase); finally, tumor resection was performed in another asleep phase. A transsylvian approach was always preferred over a transcortical one unless the opercula were involved by tumor infiltration.

Data from intraoperative monitoring and mapping were recorded and used for statistical analysis. Intraoperative MEP amplitude variations were stratified (no variation, <50% decrease, and >50%) and correlated to neurological outcomes. Similarly for awake mapping patients, the occurrence of deficit (both cognitive and sensory–motor) at the end of the procedure were correlated to neurological outcome.

Statistical Analysis. Analyses were performed using R software v4.2.2 (R Core Team, Vienna, Austria). Data distribution of continuous variables was assessed using the Shapiro–Wilk test. Continuous data are presented as median and range unless otherwise specified. Differences in continuous variables between groups were tested using the Kruskal–Wallis test with Dunn’s post hoc test. Categorical variables were described as absolute and/or relative frequencies, and Fisher’s exact test was used for categorical comparisons. Multiple logistic regression models were built to evaluate the influence of various factors on categorical outcomes, with model selection based on Bayesian Information Criterion (BIC) minimization. *p*-values < 0.05 were considered statistically significant.

## 3. Results

Forty-three patients were included in this study. The mean age was 48.2 years, and the male/female ratio was 2.31. Seizures were the most common onset symptom (51.2%), and 48.8% of patients had normal neurological examinations. The preoperative mean KPS was 92.6.

The tumor was located in the left hemisphere in 37.2% of cases, with a mean volume of 48.1 cc. Tumor classification according to the Berger–Sanai and Kawaguchi systems is shown in Table 1.

Asleep–awake–asleep anesthesia was used for seven patients (16.2%). For patients operated on via IONM (36 patients), 48.8% of cases showed no MEP decrease or neurological deficits; a >50% baseline reduction was recorded in only four cases (9.3%) (Table 2). For awake patients, 3/7 (42.9%) showed worsening at the end of the surgery.

Postoperatively, a good outcome was achieved in twenty-four patients (55.8%); only four patients (9.3%) showed definitive worsening at 6-month follow-up. The mean KPS at the last follow-up was 93. GTR and STR were achieved in 53.5% and 30.2% of patients, respectively, with a mean EOR of 92.2% and a mean residual volume of 3.35 cc. Pathology revealed high-grade glioma (HGG) in 31 cases (72.1%), with glioblastoma being the most common histotype (44.2%). Twenty-five patients (58.1%) received adjuvant chemo- and radiotherapy according to the Stupp protocol [13] (Table 3).

At the last follow-up, thirty-one cases showed disease progression (72.1%), while only one patient died.

### 3.1. Extent of Resection

A GTR was achieved in 100% of B-S zone I tumors but only in 13.33% of giant lesions (*p* = 0.001) (Figure 1). Significant differences were noted between these groups and tumors located in the B-S I, I + IV, and I + II zones. The median EOR was 100% in Kawaguchi class 3 and 4 tumors but 86.5% in class 2 tumors (*p* = 0.032) (Figure 2 and Figure 3).

When comparing our results with the literature-reported EOR rates, we found a moderate-to-strong correlation for B-S classification (Kendall’s tau = 0.502, *p* < 0.001). Kawaguchi et al.’s GTR rates showed a high, though non-significant, Spearman coefficient (r = 0.9, *p* = 0.083) (Table 4 and Table 5).

B-S classification was significantly correlated with GTR achievement (*p* < 0.001); logistic regression models indicated a 15.2-times-higher probability of GTR for B-S I + IV zones compared to giant tumors. After adjusting for sex, side, onset symptoms, WHO classification, and molecular profile, B-S I and I + IV zones remained significantly associated with higher GTR rates. Kawaguchi scoring showed a non-significant positive trend towards higher scores and better GTR outcomes (*p* = 0.097), though logistic models found a worse GTR rate for scores of 1 compared to higher classes.

### 3.2. Postoperative Neurological Outcome

B-S zone I was associated with stable or improved postoperative neurological status in 100% of cases, while B-S zones II + III were linked to neurological worsening in 75% of patients. A significant association was found between B-S classes and postoperative outcomes (*p* = 0.009). Logistic models indicated that B-S zones II + III had a 52.8% higher probability of a poor postoperative outcome compared to B-S zone I. Similarly, a significant trend towards better outcomes was observed with increasing Kawaguchi scores (*p* = 0.009), with grade 1 tumors having a 51.7% lower probability of good outcomes compared to grade 3 (Figure 4 and Figure 5). These associations were not significant at 6-month follow-up for either classification system (*p* = 0.459).

### 3.3. Ischemic Lesions

Postoperative significant DWI hyperintensity was more frequent in patients with lower Kawaguchi scores (*p* = 0.014). A logistic model revealed that a score of 1 increased the risk of postoperative DWI alterations by 22.5 times compared to a score of 3. In contrast, no correlation was found with B-S zones (*p* = 0.187) (Table 6).

### 3.4. IONM Changes and Clinical Outcomes

In total, 62.1% of patients without IONM changes showed a good postoperative outcome; on the contrary, it was present in 27.3% of cases with IONM worsening (*p* = 0.077). At 6-month follow-up, no difference emerged (*p* = 0.87) The Berger–Sanai classification did not emerge as a prognostic factor for clinical outcomes based on IONM changes, both immediately postoperatively and at 6-month follow-up. However, the Kawaguchi system appeared to have prognostic value; specifically, it played a role in predicting an 83.1% worse immediate postoperative outcome for grade 2 tumors compared to grade 3 (*p* = 0.084) (Table 7).

## 4. Discussion

Insular gliomas are distinct entities from both biological and clinical standpoints. In a detailed molecular analysis, Goze et al. compared pure insular gliomas with “para-limbic” gliomas, which, according to the Yasargil classification, extend into fronto-temporal and/or temporopolar regions, with or without the involvement of mesiotemporal structures [14,15]. This study found a higher incidence of IDH1/IDH2 mutations in insular gliomas, associated with smaller tumor size at onset and lower malignancy. Although the “triple-negative” subtype was identified in insular gliomas, no significant differences were observed between the two groups [16,17,18]. Insular gliomas typically follow four patterns of progression: they may remain confined to the insular cortex or extend through the anterior, inferior, or superior limiting sulci, each with specific surgical and clinical implications. For instance, spread through the superior sulcus limitans can hinder gross total resection (GTR) due to the risk of damaging the posterior limb of the internal capsule and long insular arteries (LIAs), which supply the corona radiata [6,7,13]. Tamura et al. emphasized that LIA strokes can cause neurological deficits similar to those caused by lenticulostriate arteries (LSAs), underscoring the need to preserve these vessels [16]. In 2014, Kawaguchi et al. introduced a scoring system that includes factors such as contrast enhancement, LSA involvement, tumor edge morphology, and superior sulcus limitans invasion. This 0–4 score system has clinical and surgical implications; higher scores correspond to higher resection rates and lower neurological complication [7]. In the well-known Berger–Sanai classification, tumors in zones II and III are less amenable to GTR due to the increased risk of complications like hemiparesis or hemiplegia [11]. While the Kawaguchi system emphasizes tumor characteristics and proximity to critical structures, the Berger–Sanai classification is more focused on tumor topography, which has indirect surgical and clinical consequences. In this study, we applied both the Berger–Sanai and Kawaguchi classifications to a series of insular glioma patients treated at our center. We achieved radical resection in 53.5% of cases, which is relatively high compared to other series [19,20,21]. Although some recommend a transopercular approach to minimize vessel manipulation and reduce motor deficits [22,23,24,25,26], we favored a transylvian approach for its more natural route and improved control of vessels within the cistern. This approach also provided an anatomic perspective, using the M1 and lateral lenticulostriate arteries as depth markers for the surgical cavity. Resection rates varied depending on the classification system used. Berger–Sanai giant tumors had the lowest extent of resection (EOR), as expected, and zone II + III involvement significantly reduced GTR and subtotal resection (STR) rates, aligning with previous studies. For example, while Hervey-Jumper et al. reported a median EOR of 75.5% for zone II tumors [8], Sufianov et al. achieved an EOR of over 90%. In our series, we did not encounter any pure zone II tumors, but we agree that proximity to the corticospinal tract and LIA involvement can limit resection [27]. Using the Kawaguchi scoring system, higher scores were associated with higher EOR. Logistic regression showed that grade 1 tumors had a 90.1% lower GTR rate compared to grade 3, though no significant difference was observed between grade 3 and grade 4 tumors. Further analysis revealed that the Berger–Sanai classification had moderate-to-strong predictive power for EOR, consistent with findings from Hervey-Jumper et al. [8]. Although the Kawaguchi system did not show strong predictive ability, a positive trend was evident, supported by a high Spearman coefficient. The Berger–Sanai classification also significantly predicted GTR, particularly for tumors in zones I and IV, while Kawaguchi grade 1 was associated with a lower GTR rate. Interestingly, histotype did not affect EOR rates, suggesting that maximal safe resection should be pursued even if a high-grade glioma is suspected. Postoperative neurological deterioration occurred in 44.2% of cases, comparable to the 45.5% reported by Biswas et al. (30 out of 66 patients) [19]. This may be related to ischemic damage or surgical manipulation, as postoperative MRI revealed significant DWI hyperintensity in 32.6% of cases. Moreover, the high incidence of glioblastoma we observed in this series could be a notable factor, as higher-grade tumors may be more likely to involve LSAs, increasing the risk of postoperative neurological worsening. Nevertheless, statistical analysis did not identify it as a confounding factor. Both the Berger–Sanai and Kawaguchi systems were linked to neurological outcomes. Tumors in Berger–Sanai zones II + III were 52.8% more likely to cause immediate postoperative complications than those in zone I. Interestingly, giant tumors were less risky than those in zones II + III, possibly due to the original definition of “giant” tumors. According to Rossi et al., LSA infiltration by the tumor is the only factor associated with postoperative neurological deterioration, and these vessels are easier to control surgically than LIAs [28]. These findings underscore the importance of Kawaguchi scores in predicting postoperative outcomes, as confirmed by logistic regression models that also linked the Kawaguchi system with DWI changes. At the 6-month follow-up, 9.3% of patients (four cases) showed permanent neurological decline. No significant correlation was found with either classification system, possibly due to the small sample size. Alternatively, postoperative DWI hyperintensities may not always reflect irreversible ischemic damage, or recovery from ischemia may still be possible. Hou et al. reported that while 58.6% of patients had significant DWI hyperintensities postoperatively, only 14.6% were related to core ischemia in the corona radiata or posterior limb of the internal capsule [29]. Intraoperative neurophysiological monitoring (IONM) is a well-established tool for detecting impending stroke during glioma surgery and allows for a higher resection rate [30]. Motor-evoked potentials (MEPs) are more sensitive and specific than somatosensory-evoked potentials (SSEPs), with a >50% reduction in MEP amplitude strongly linked to paralysis risk. In our study, we observed a positive trend toward better outcomes among patients without intraoperative changes (OR 4.20), but it did not reach statistical significance, likely due to our study’s small sample size. Regarding classification systems, the Kawaguchi classification correlated with IONM changes, with lower-grade tumors showing higher rates of significant MEP reductions. This may reflect the Kawaguchi system’s focus on the tumor’s relationship to LSAs and the superior extremity of the circular sulcus, making it more directly connected to vessel damage than the Berger–Sanai classification. Our work has two primary limitations. Firstly, the sample size is small, limiting the strength of our conclusions; a multicenter study would be beneficial to increase the sample size. Secondly, the retrospective nature of this study could have introduced selection and data biases.

## 5. Conclusions

The Berger–Sanai and Kawaguchi classifications are two distinct approaches for categorizing tumors and predicting outcomes in insular glioma surgery. In this study, we compared their performance across a series of consecutive cases. Both classifications are linked to the extent of resection, but the Berger–Sanai system is a more reliable predictor of GTR, particularly for tumors in zones I and IV. Both systems are also effective in predicting postoperative neurological deterioration, although this correlation does not hold at the 6-month follow-up. Notably, lower Kawaguchi scores are associated with a higher risk of postoperative DWI hyperintensities, >50% reductions in MEP amplitude, and neurological deficits at the end of surgery, whereas the Berger–Sanai system has less impact on these outcomes. In conclusion, our findings support the hypothesis that these two scales offer different predictive strengths. Future efforts should aim to create a multidimensional, comprehensive classification system that integrates the strongest predictors from both scales to improve prognostic accuracy for neurological and oncological outcomes.

## Figures and Tables

**Figure 1 cancers-16-04137-f001:**
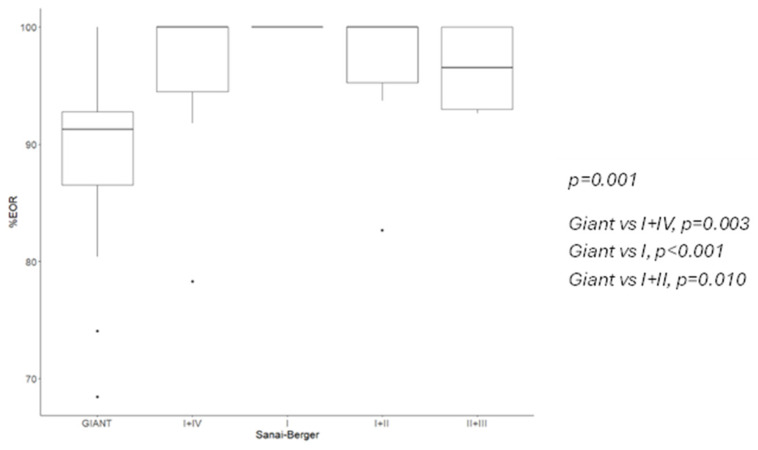
EOR according to B-S classification, median and 95% CI.

**Figure 2 cancers-16-04137-f002:**
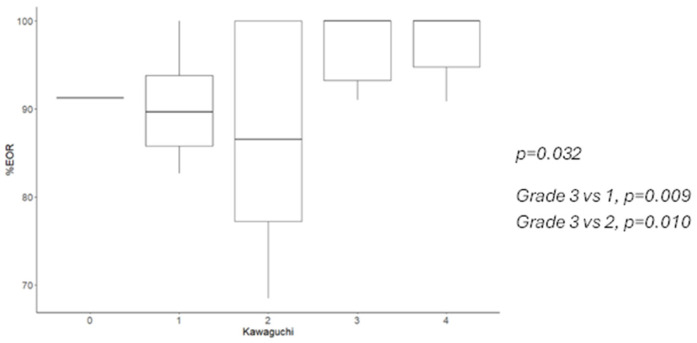
EOR according to Kawaguchi classification, median and 95% CI.

**Figure 3 cancers-16-04137-f003:**
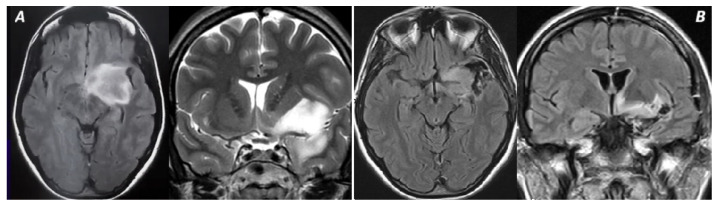
In this picture, a Kawaguchi grade 2 tumor is represented. (**A**) The invasion of LSAs and anterior perforated substance lead to the impossibility of performing a GTR, as shown in (**B**).

**Figure 4 cancers-16-04137-f004:**
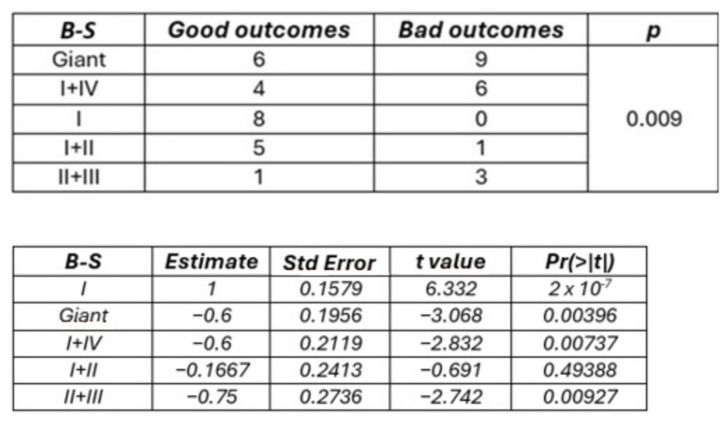
Correlation between B-S Classification and Immediate postoperative outcome according to Fisher’s test (**Upper**) and logistic model (**Lower**). Class I is the intercept.

**Figure 5 cancers-16-04137-f005:**
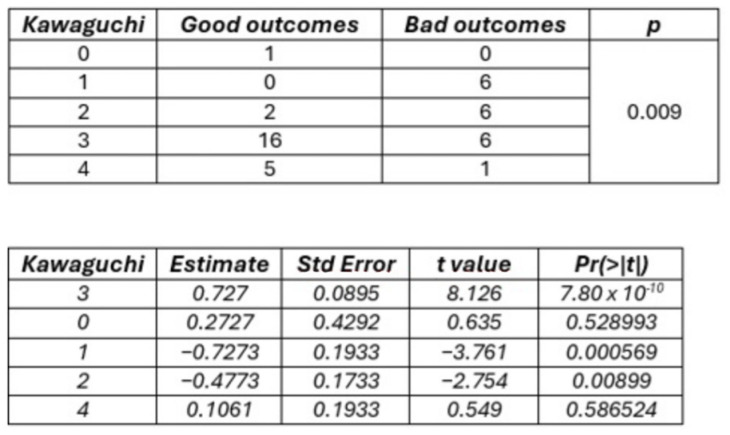
Correlation between Kawaguchi classification and immediate postoperative outcome according to Fisher’s test (**Upper**) and logistic model (**Lower**). Grade 3 is the intercept.

**Table 1 cancers-16-04137-t001:** Preoperative data.

**Age**	47 Years
**M/F**	2.31
**Preoperative KPS**	92.6
**Mean Tumor Volume**	48.09 cc
**Tumor side**	
Left	16
Right	27
**Onset**	
Sensory–motor deficit	4
Aphasia	4
Cognitive impairment	4
Seizures	21
Incidental	2
Relapse	1
Combined symptoms	6
**Neurological Examination**	
Sensory–motor deficit	6
Aphasia	4
Cognitive impairment	9
Normal	21
Others	3
**Berger–Sanai**	
Giant	15
I + IV	10
I	8
I + II	6
II + III	4
**Kawaguchi**	
0	1
1	6
2	8
3	22
4	6

**Table 2 cancers-16-04137-t002:** Intraoperative data.

**Anesth Tech**	
**Awake**	7
**Asleep**	36
**IONM**	
No EP alteration nor deficit	29
>50% EP decrease	4
<50% EP decrease	3
Deficits at the end	4

**Table 3 cancers-16-04137-t003:** Postoperative data.

**Immediate Postop Cond**	
**Stable**	22
**Improved**	2
**Worse**	19
**WHO**	
2	12
3	12
4	19
**Histotype**	
Oligodendroglioma	8
Astrocitoma	16
Glioblastoma	19
**EOR**	
GTR	23
STR	13
PTR	7
**Adjuvant therapy**	
W/S	4
RT	13
Chemo	1
RT + Chemo	25
**6-month follow-up**	
Stable	21
Completely improved	11
Partially improved	7
Worsened or def worsened	4
**Definitive KPS**	93
**Mean FU**	38.04 m
**Disease at last FU**	
Progression	31
Stable	11
Death	1

**Table 4 cancers-16-04137-t004:** Ability of B-S classification to predict median EOR. Exp: expected.

B-S	EOR (%) (Range)	Exp EOR (%)	*p*	Kendall τ (CI95%, *p*)
I	100 (100–100)	92.0	0.008	0.502 (0.369, 0.636) *p* < 0.001
I + II	100 (82.7–100)	78.9	0.031	
I + IV	100 (78.3–100)	78.0	0.002	
II + III	96.6 (92.6–100)	84.0	0.125	
Giant	91.2 (68.4–100)	76.4	0.007	

**Table 5 cancers-16-04137-t005:** Ability of Kawaguchi classification to predict GTR.

Kawaguchi	GTR (%) (Cases)	Exp GTR (%)	Spearman Coefficient (r, *p*)
0	0 (0)	0	0.9, 0.083
1	16.7 (1)	11.8	
2	37.5 (3)	39.3	
3	68.2 (15)	47.4	
4	66.7 (4)	100	

**Table 6 cancers-16-04137-t006:** Correlation between Kawaguchi classification and significant postoperative DWI hyperintensity.

Kawaguchi	DWI Hyper	Normal	*p*
0	0	1	0.014
1	5	1	
2	4	4	
3	4	18	
4	1	5	

**Table 7 cancers-16-04137-t007:** Logistic model showing the influence of IONM on postoperative outcome according to Kawaguchi classification. Kawaguchi 3 is the intercept.

	Estimate	Std Error	t Value	Pr (>|t|)
Kawaguchi 3	1.1934	0.5531	2.158	0.031
Kawaguchi 0	17.3727	6522.639	0.003	0.9979
Kawaguchi 1	−19.4805	2601.66	−0.007	0.994
Kawaguchi 2	−1.7818	1.0323	−1.726	0.0844
Kawaguchi 4	0.4824	1.4261	0.338	0.7351
>50% MEP decrease	−0.7936	1.4656	−0.542	0.5881
<50% MEP decrease	−1.1934	1.5185	−0.786	0.4319
End procedure deficit	−1.7841	1.4133	−1.262	0.2068

## Data Availability

Data are available on request to the corresponding author.

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
