# Peer review of "Predicting Extent of Resection and Neurological Outcome for Insular Gliomas: An Analysis of Two Available Classifications"

_cancers, 2024, doi:10.3390/cancers16244137_

Round 1

Reviewer 1 Report

Comments and Suggestions for Authors

This study retrospectively examines the predictive indicators for resection and complications in surgical excision of insular gliomas, to identify more effective measures compared to existing classifications. Given the different perspectives emphasized by the BS and Kawaguchi classifications, I agree that the study's focus on finding a more effective indicator is meaningful. However, compared to the original classification studies, the BS classification analyzed 104 cases, while the Kawaguchi classification analyzed 83 cases, both of which are significant case numbers for tumors in this region. Furthermore, in terms of the proportion of GBM cases, the BS classification had only 10/115 cases (8.7%), and the Kawaguchi classification had 31/83 cases (37%), whereas this study includes a high proportion of GBM at 19/43 cases (44%), which presents an issue. Comments are outlined below:

  1. Regarding the inclusion of both GBM and non-GBM cases
    The BS classification attempted to exclude GBM cases in its analysis, given that GBM influences treatment outcomes and resection feasibility. For insular GBM involving the lenticulostriate arteries (LSAs), ischemia in the LSA region is more likely to occur. In addition, for GBM, tumor size correlates with surgical difficulty. Given the high proportion of GBM cases in this study, it is necessary to address the impact this may have on data interpretation and discussion.
  2. Inclusion of Kaplan-Meier analysis and resection rates in Kawaguchi classification
    The Kawaguchi classification also incorporates prognosis analysis using Kaplan-Meier and evaluates cases where over 90% subtotal removal was achieved. In contrast, this study shows an 84% subtotal resection or higher, presenting a notable difference in profiles compared to both classifications. Due to the limited sample size and the small number of cases meeting pathological diagnosis, resection rate, and individual classification criteria, I do not believe sufficient analysis has been achieved to draw definitive conclusions.

Author Response

  1. Regarding the inclusion of both GBM and non-GBM cases
    The BS classification attempted to exclude GBM cases in its analysis, given that GBM influences treatment outcomes and resection feasibility. For insular GBM involving the lenticulostriate arteries (LSAs), ischemia in the LSA region is more likely to occur. In addition, for GBM, tumor size correlates with surgical difficulty. Given the high proportion of GBM cases in this study, it is necessary to address the impact this may have on data interpretation and discussion.
  1. Thank you for your observations. This is an important point to elaborate. We will deepen and stress this concept in discussion paragraph.
  1. Inclusion of Kaplan-Meier analysis and resection rates in Kawaguchi classification
    The Kawaguchi classification also incorporates prognosis analysis using Kaplan-Meier and evaluates cases where over 90% subtotal removal was achieved. In contrast, this study shows an 84% subtotal resection or higher, presenting a notable difference in profiles compared to both classifications. Due to the limited sample size and the small number of cases meeting pathological diagnosis, resection rate, and individual classification criteria, I do not believe sufficient analysis has been achieved to draw definitive conclusions.

The small sample size is a limit of our study, undoubtely. We reached some significant results but the low number of patients did not permit us to reach strong conclusions. We added these considerations to our paper. 

Reviewer 2 Report

Comments and Suggestions for Authors

Thank you very much to give me an opportunity to review this manuscript.

Authors demonstrated that Kawaguchi classification including LSAs involvement and superior sulcus limiting invasion predicted extent of resection (EOR) and postoperative neurological deficits. And I am impressed excellent surgical results of EOR and clinical outcome in the present cohort. I agree to authors’ conclusions and suggestions.

#1. As authors described, Sanai-Berger and Kawaguchi classification system might reflect resection rate and short-term postoperative clinical outcome. How was long-term clinical outcome of patients in the present cohort of both lower and high-grade gliomas in the insula? Did authors insisted that extent of resection should contribute prolongation of progression free survival (PFS) and overall survival (OS)?

#2. IDH-1 status should be important to determine clinical outcome. Even with sophisticated surgical manipulation with intraoperative neurophysiological monitoring (IONM), it should be extremely difficult to achieve gross total resection (GTR) of insula gliomas, especially when involving LSAs or eloquent area such as Broca, corona radiata, and internal capsule. Generally speaking, clinical outcome of PFS and OS for IDH-1 mutated glioma is supposed to be favorable compared with IDH-1 wild type. therefore, I propose “two-staged surgery” with confirmation of genetic status including IDH-1 and 1p19q codeletion at the initial surgery. Personally, I think that authors described additionally in the discussion section.

#3. I strongly recommended that illustrative case should be presented with neuroimage of pre- and post-operative MRI, suggestive of usefulness for Kawaguchi classification.

Minor points

#1. Please spell out “IONM” in the simple summary.

#2. (line 33) The median extent of resection (EOR) was not 100%, but 92.2% as described in the result section (line 138). Please correct it.

Author Response

Authors demonstrated that Kawaguchi classification including LSAs involvement and superior sulcus limiting invasion predicted extent of resection (EOR) and postoperative neurological deficits. And I am impressed excellent surgical results of EOR and clinical outcome in the present cohort. I agree to authors’ conclusions and suggestions.

#1. As authors described, Sanai-Berger and Kawaguchi classification system might reflect resection rate and short-term postoperative clinical outcome. How was long-term clinical outcome of patients in the present cohort of both lower and high-grade gliomas in the insula? Did authors insisted that extent of resection should contribute prolongation of progression free survival (PFS) and overall survival (OS)?

  1. Thank you very much for your observations. Our works aimed to evaluate the prognostic role of the two classifications in terms of EOR, clinical outcomes and other issues. Maximal safe resection is a well-known factor influencing PFS and OS, as long as no neurological worsening appears. As many papers demonstrated the crucial role of maximal safe resection on glioma prognosis, we would have risked to fournish redundant informations by remarking it. On the contrary, we analyzed long term prognosis in terms of clinical status and we did not find differences between the two classifications.

#2. IDH-1 status should be important to determine clinical outcome. Even with sophisticated surgical manipulation with intraoperative neurophysiological monitoring (IONM), it should be extremely difficult to achieve gross total resection (GTR) of insula gliomas, especially when involving LSAs or eloquent area such as Broca, corona radiata, and internal capsule. Generally speaking, clinical outcome of PFS and OS for IDH-1 mutated glioma is supposed to be favorable compared with IDH-1 wild type. therefore, I propose “two-staged surgery” with confirmation of genetic status including IDH-1 and 1p19q codeletion at the initial surgery. Personally, I think that authors described additionally in the discussion section.

  1. Your proposal is interesting, but the role of maximal safe resection remains important even in case of IDHwt or 1p19q codeleted tumors. Anyway, we will stress this concept in Discussion paragraph.

#3. I strongly recommended that illustrative case should be presented with neuroimage of pre- and post-operative MRI, suggestive of usefulness for Kawaguchi classification.

  1. We will add it.

Minor points

#1. Please spell out “IONM” in the simple summary.

#2. (line 33) The median extent of resection (EOR) was not 100%, but 92.2% as described in the result section (line 138). Please correct it.

  1. We will fix it.

Reviewer 3 Report

Comments and Suggestions for Authors

The authors are to be commended for their excellent results in treating patients with this challenging pathological entity and the evaluation of their treatment as it relates to reported classification systems.

There are a number of revisions that should be made to the presentation which would make the paper have more impact.

1. There are a number of typographical errors, which I hope will be addressed in type setting. The proof included hyphens in the wrong place  eg line 2 in-sular, line 18 to-pography and throughout. Whilst this may be necessary for the publication size in the text it should not be in the title.

2. In most of the literature the first classification system is referred to as the Berger-Sanai classification rather than Sanai-Berger

3. the sentence line 19 needs revising  -it is not clear what 'one glioma attitudes' means.

4. In the abstract line 33. the sentence  'the median extent of resection (EOR) was 100%.' This statement needs qualifying as either it relates to a single subtype eg zone 1 tumours line 146 or refers to all tumours in which radical resection was achieved in 53.5% of cases line 227.

5. The two classification systems are described in lines 61 - 66. The significance of the two classification systems could benefit from better description, ideally with the use of figures. As currently described it requires the reader to access the papers themselves to understand the current study.

6. The Kawagachi system specifies four criteria which could be made clearer line 66.

7. A higher score in the Kawagachi system predicts a higher extent of resection.line 67 - 

8. Male to female ratio is usually presented as numbers eg M:F 30:13  or 2.3:1 -  line 123.

9. line 130 - most awake craniotomy surgery is performed asleep-awake-asleep with the patient awake for the resection of the tumour - please clarify

10. It would be useful to know the extent of resection as compared to pathological subtype if this information is available - does it influence EOR? what is the significance of the tumour type?

Comments on the Quality of English Language

minor changes as above

Author Response

The authors are to be commended for their excellent results in treating patients with this challenging pathological entity and the evaluation of their treatment as it relates to reported classification systems.

There are a number of revisions that should be made to the presentation which would make the paper have more impact.

  1. There are a number of typographical errors, which I hope will be addressed in type setting. The proof included hyphens in the wrong place  eg line 2 in-sular, line 18 to-pography and throughout. Whilst this may be necessary for the publication size in the text it should not be in the title.
  2. We will fix it
  3. In most of the literature the first classification system is referred to as the Berger-Sanai classification rather than Sanai-Berger
  4. We will fix it
  5. the sentence line 19 needs revising  -it is not clear what 'one glioma attitudes' means.
  6. We will clarify the concept.
  7. In the abstract line 33. the sentence  'the median extent of resection (EOR) was 100%.' This statement needs qualifying as either it relates to a single subtype eg zone 1 tumours line 146 or refers to all tumours in which radical resection was achieved in 53.5% of cases line 227.
  8. It is a type error. The mean extent was lower and we will fix the error.
  9. The two classification systems are described in lines 61 - 66. The significance of the two classification systems could benefit from better description, ideally with the use of figures. As currently described it requires the reader to access the papers themselves to understand the current study.
  10. We explained factors related to two classifications. Anyway, references of the two classifications can be found in the right paragraph.
  11. The Kawagachi system specifies four criteria which could be made clearer line 66.
  12. We will specify them.
  13. A higher score in the Kawagachi system predicts a higher extent of resection.line 67 – 
  14. Male to female ratio is usually presented as numbers eg M:F 30:13  or 2.3:1 -  line 123.
  15. We will fix it.
  16. line 130 - most awake craniotomy surgery is performed asleep-awake-asleep with the patient awake for the resection of the tumour - please clarify.
  17. We will explain the technique in “M&M” paragraph.
  18. It would be useful to know the extent of resection as compared to pathological subtype if this information is available - does it influence EOR? what is the significance of the tumour type?
  19. Requested analysis was already done. Anyway, we will specify it.

Round 2

Reviewer 3 Report

Comments and Suggestions for Authors

Thank you for the revision- a couple of minor typographical changes.

line 67-68 should read ' higher scores in this system are associated with a greater extent of resection.

line 136 - should read 'asleep-awake-asleep'

Author Response

Thank you for the revision- a couple of minor typographical changes.

line 67-68 should read ' higher scores in this system are associated with a greater extent of resection.

line 136 - should read 'asleep-awake-asleep'

  1. We fixed them.